# A Fast Calculation Model for Local Head Loss of Non-Darcian Flow in Flexural Crack

**Jian Liu, Chenya Mou, Kai Song, Peng Luo, Liang He and Xue Bai \***

Faculty of Geosciences and Environmental Engineering, Southwest Jiaotong University, Chengdu 610031, China; liukai-102@163.com (J.L.); mouchenya@163.com (C.M.); songkailw@163.com (K.S.); 18380444501@139.com (P.L.); lzheliang@163.com (L.H.)

\* Correspondence: snooww@my.swjtu.edu.cn; Tel.: +86-150-082-511-34

**Abstract:** Local head loss caused by fracture intersection is often ignored because there has not been a simple method to calculate it until now. Relevant research shows that neglecting the local flow resistance leads to inaccurate results, especially when the velocity and cross angle are large. Therefore, it is necessary to find a portable method for calculation. Physical experiments of single fracture with different apertures (e = 0.77, 1.18, 1.97, 2.73 mm) were set up first to study the flow characteristics, showing obvious non-Darcian flow, which can be depicted by the Forchheimer equation when the flow velocity is sufficiently large. The computational fluid dynamics (CFD) software ANSYS FLUENT was used to build numeric simulation models. A good correlation between CFD simulation results and physical experiment results was found (Pearson's correlation coefficient > 0.99). Then, the CFD models of flexural crack with different angles from 30° to 150° were established to compute the pressure drop of flexural crack at different velocity. It was found that the local head loss of the flexural crack varied with the bending angle, and its coefficient was expressed by the deformation of the logistic equation. By using this model, as well as a frictional head loss equation fitted by Forchheimer equation, the head loss of crossed fissures with fixed fracture aperture could be easily calculated.

**Keywords:** non-Darcian flow; fracture aperture; fracture shape; local head loss

## 1. Introduction

Fracture fluid in nature usually moves at a high speed. When its velocity reaches a certain value, the loss of fluid velocity is dominated by the inertial force, and this phenomenon is called the high speed non-Darcian seepage [1]. Basak et al. [2] and Madhav found that high speed non-Darcian flow is prevalent in hydraulic engineering, whereas Tartakovsky et al. [3] observed the same result in confined aquifers. Research on the law of non-Darcian seepage of high-speed fluids forms the basis of the effective prediction of complex fracture seepage in rock masses and is key to ensuring safety and ecological stability in engineering construction [4]. To describe the non-Darcian fluid in fissures, Zimmerman et al. [5] established a single sandstone fissure to determine the Reynolds number boundary for the best characterization equation of non-Darcian flow. Qian et al. [6] studied the evolution of flow from Darcian to non-Darcian in fissures under confined and unconfined condition and concluded that the critical Reynolds number decreases gradually with the increase in fissure width. Quinn et al. [7] carried out an experimental study on non-Darcian flow in fissures of a dolomite aquifer and quantitatively analyzed its hydraulic gradient and velocity.

At the microscopic scale, the Navier–Stokes equation is often used to study the relationship between the velocity and pressure of Newtonian fluids in cracks. Because of its complex calculation, however, it is difficult to apply it to engineering practice. To improve the efficiency of engineering

applications, a quadratic equation ($J = Av + Bv^2$) and an exponential equation ($J = Cv^{-m}$) are used to under limited conditions [8–11].

The Forchheimer equation can express the characteristics of non-Darcian flow in fracture [12–15]:

$$-\nabla P = aQ + bQ^2 \qquad (1)$$

where Q is volumetric flow velocity, a and b are model coefficients, and $-\nabla P$ is the pressure gradient.

The factors affecting the flow of fissure fluid are complex and include the shape of the fissure, degree of contact, roughness, fluid viscosity, and external pressure [16–19]. Liu et al. [20] studied the law of fluid movement in a fracture network and found that the coefficient (a, b) of the Forchheimer equation decreases with an increase in the fracture aperture, and the error in model fitting can be reduced by expressing the coefficients of the Forchheimer equation as a power equation of fracture aperture [14]. Olson et al. [21] studied the effect of the ratio of the fracture opening to its height on the flow pattern, and Shu et al. [22] studied the variation in head loss with the width of the fissure and the velocity of flow when water flows through L-shaped fissures. Li et al. [23] studied the correlation between the head loss of a fluid in cross fissures and the width as well as roughness of the fissures.

This study proposes a portable calculation model where the aperture and the shape of flexural crack are considered to predict head loss directly. It is difficult to consider all factors when studying the characteristics of flow in fractures. In order to investigate the influence of each one, the synergistic effects of each condition need to be reduced, and the influence rules of each condition need to be researched separately. Fracture aperture and shape are two of the important parameters to generalize fracture development, which explains why only the two most significant parameters were considered here.

## 2. Experiments and Methodology

The fracture aperture was first used as the control condition to construct an indoor physical model, and head loss and velocity were observed to analyze the characteristics of the fluid in the single fissure. Following the verification of the computational fluid dynamics (CFD) model in simulation of characteristics of the fluid in the single fissure with parameters most used [24–28], the relationship between the shape of the fracture (corner θ) and head loss was studied using it.

### 2.1. Laboratory Test Device

The test device shown in Figure 1 consists of a water intake system, fissure system, pressure measurement system, and width measurement system. The water intake system was a set of water distribution devices that controlled the water intake head and ensured the stability of water inflow by setting an overflow port on the water distributor. The main fractured rock mass of the fractured system was composed of marble purchased from the stone market ($50 \times 30 \times 1.8$ cm), and its width could be adjusted to 0.77, 1.18, 1.97, and e = 2.73 mm, respectively, during the experiment. The piezometric system was composed of piezometric tubes and observation plates buried in different sections. The measurement system consisted of a device to measure the width of the fracture aperture and a flow-measuring device. The crack photos and ruler were taken by a 1080P HDMI digital microscope are magnified and read on the computer, and then the gap width of each section was accurately measured by using the image recognition technology of the supporting software, whose accuracy can be controlled at 0.01 mm. The flow-measuring device used a stopwatch, 1000 mL or 2000 mL beaker, and an electronic balance to determine the flow velocity. The accuracy of the electronic balance was 0.01 g. The climate did not change significantly, and the water temperature fluctuated slightly during the test, around 12 °C.

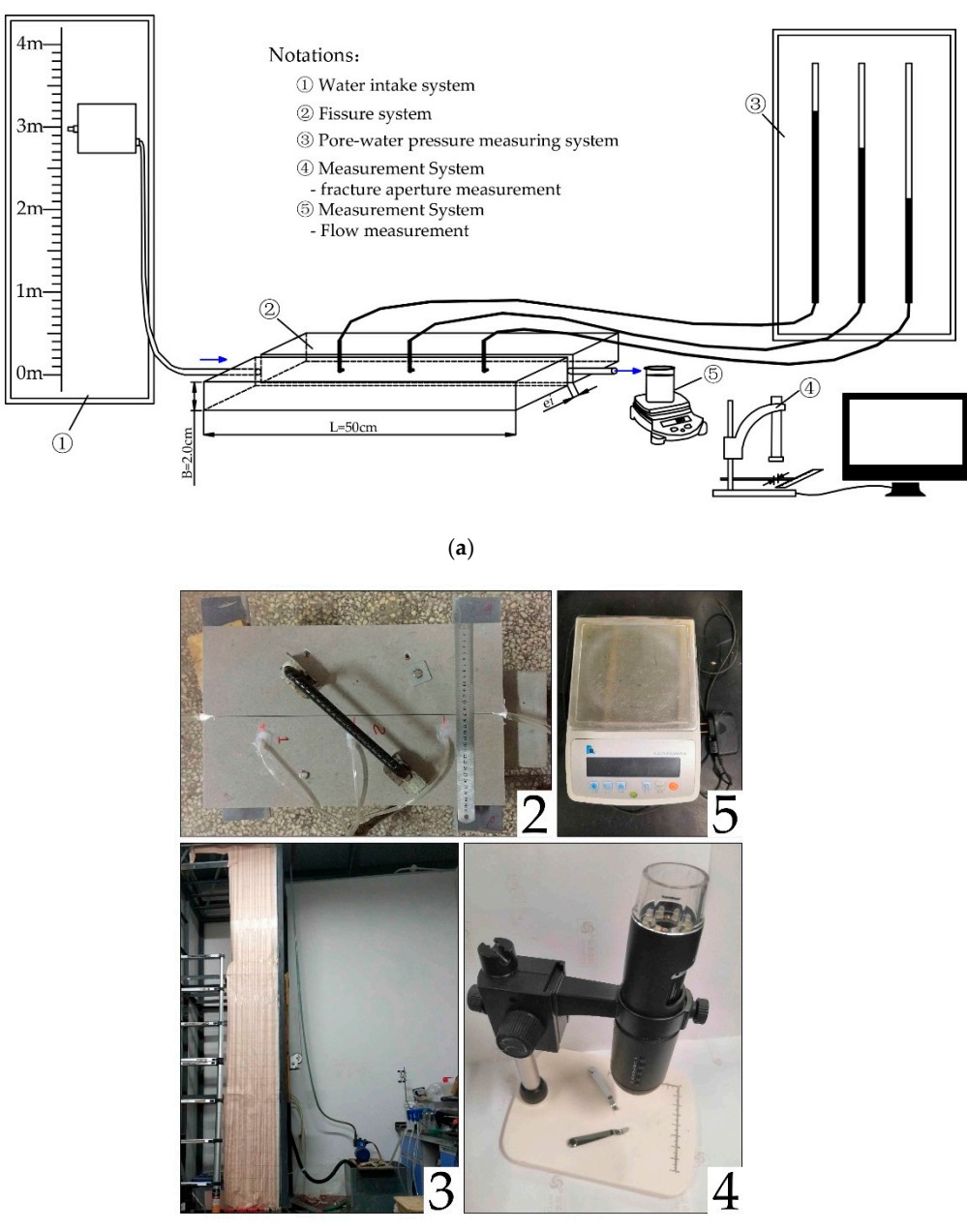

(a)

(b)

**Figure 1.** The experimental model. (**a**) Conceptual model; (**b**) Physical model. ① is the water intake system, which can control the water intake head and ensure the stability of water inflow; ② is fissure system, of which the fracture width can be adjusted to 0.77, 1.18, 1.97, and e = 2.73 mm; ③ is the piezometric system; ④ is fracture aperture measurement system, the major instrument is a 1080P HDMI digital microscope; ⑤ is flow measurement system.

*2.2. Laboratory Test Method*

Taking the fracture aperture and water pressure as control conditions, the system was reorganized when the fracture aperture was changed. The construction and operation of the device were as follows:

1.  Assembling: Once the stone had been cut into two pieces in a professional stone crushing workshop, we measured and arranged it in the laboratory and installed the inlet and outlet pipe as well as pressure tubes. The fracture aperture was measured after some necessary adjustment and fixing of the stones, and then all of the interface was sealed with acrylic glue. When the test

was complete, the sealant was cleaned and the width of the gap readjusted to another one. The stones were then fixed and sealed again. We planned to set the gap widths within the range of 0.05–0.10, 0.10–0,15, 0.15–0.2, and 0.2–0.3 cm, while the actual measured gap widths were 0.77, 1.18, 1.97, 2.73 mm, respectively.

2.  Connection: After checking the sealing of the cracked plate, the water inlet was connected to the water intake system, and the pressure gauge pipe was connected to the pressure measurement system. It was ensured that no bubbles were in the pressure gauge pipe during the experiments.

3.  Test: Water pressure was adjusted through moving the height of the water tank, and more than 5 min was needed to wait for piezometric head stabilization. The fluid flow was measured by the volumetric method mentioned previously. The range of test conditions is shown in Table 1.

**Table 1.** Range of pressure losses and water flow rate

| Experimental Results | Fracture Aperture | | | |
|---|---|---|---|---|
| | **0.077 cm** | **0.118 cm** | **0.197 cm** | **0.273 cm** |
| Range of water head loss (cm) | 6–92 | 6–80 | 3–50 | 2–20 |
| Range of water flow Q (mL/s) | 2–26 | 8–50 | 19–93 | 26–97 |

*2.3. Numeric Simulation Model*

CFD numerical calculation module can be used to simulate the laminar and turbulent flows of Newtonian and non-Newtonian fluids in hydraulics. Thakur et al. [29] used CFD simulations to study the influence of the size of Newtonian and non-Newtonian fluids on their flow fields and power consumption, Aubin et al. [30] modeled and analyzed turbulence in a stirred tank based on a CFD model, and Apsley et al. [31] used a CFD model to study the effects of roughness on turbulent flow. ANSYS FLUENT 19.1 (ANSYS, Canonsburg, PA, USA) was employed to construct a CFD model to simulate the characteristics of flow in fractures in this study.

2.3.1. Conceptual Model

A single fracture model with length of 100 mm, height of 20 mm, and aperture of 1 mm was built in FLUENT, while the bending angle of flexural crack was set to 0, 30, 45, 60, 90, 120, and 135 degrees. Because the viscous force had little influence on the characteristics of motion of high speed flow, the model was set up based on turbulent state of an ideal fluid, and the velocity at the inlet as well as a pressure of zero at the outlet were set as boundary conditions. The structure of the numerical model is shown in Figure 2.

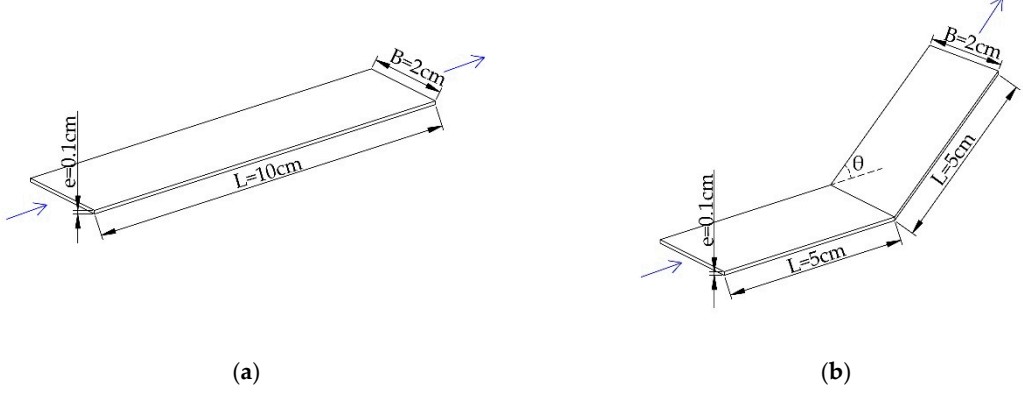

(**a**)                                                                                      (**b**)

**Figure 2.** Structure of the fissures. (**a**) Single fracture or straight fracture. (**b**) Flexural crack with θ of 30, 45, 60, 90, 120, and 135 degrees.

### 2.3.2. Mathematical Model

The CFD model was established based on the size and parameters of the physical model. High speed non-Darcian flow in the field of physical experiments was simulated by using the realizable k–ε model, which was proposed by Launder and Spalding in 1972 and subsequently constructed by Shih et al. [32] in 1995. Based on the time-averaged continuity equation, the equations of turbulent dynamic transport and the turbulence dissipation rating were established.

Turbulence dynamic transport equation:

$$\varrho \frac{d_k}{d_t} = \frac{\partial}{\partial x_i}\left[\left(\mu + \frac{\mu_t}{\sigma_k}\right)\frac{\partial k}{\partial x_i}\right] + G_k - \varrho\varepsilon \tag{2}$$

Transport equation of turbulence dissipation rate:

$$\varrho \frac{d_\varepsilon}{d_t} = \frac{\partial}{\partial x_i}\left[\left(\mu + \frac{\mu_t}{\sigma_\varepsilon}\right)\frac{\partial \varepsilon}{\partial x_i}\right] + \varrho C_1 S\varepsilon - \varrho C_2 \frac{\varepsilon^2}{k + \sqrt{v\varepsilon}} \tag{3}$$

where k represents the dynamics of turbulence, ε is its dissipation rating; t is time; ϱ is the density of the fluid; $v$ is velocity; μ is dynamic viscosity coefficient, since the experimental temperature is 12 °C, μ = 1.2362; $\mu_t$ is turbulent dynamic viscosity; $G_k$ is the generation term of turbulent kinetic energy k due to the mean velocity gradient; S is the deformation velocity tensor; $x_i$ is coordinate component; and $\omega$ is the constant angular velocity. The values of $\sigma_k$ and $\sigma_\varepsilon$ were 1 and 1.2, respectively. They represent Prandtl numbers corresponding to turbulent kinetic energy k and ε, respectively. The value of $C_2$ was 1.9, whereas $C_1$, $\mu_t$ can be defined as:

$$C_1 = \max\left[0.43, \frac{\eta}{\eta + 5}\right] \tag{4}$$

$$\mu_t = \varrho C_\mu \frac{k^2}{\varepsilon} \tag{5}$$

$C_\mu$ represents model coefficients, the experimental as well as DNS data on the inertial sublayer of a channel or boundary layer flow suggest that $C_\mu$ = 0.09. The values of the parameters were recommended and by Launder et al. [33,34].

## 3. Results and Discussion

### 3.1. Data Analysis of Laboratory Physical Model Test

In the physical experiment of single fracture, aperture e (0.77, 1.18, 1.97, 2.73 mm) was used as the test variable. The pressure gradient −∇P corresponding to different flow Q under different fracture apertures was obtained and is analyzed in Table 2 and plotted in Figure 3.

**Table 2.** Regression of the linear and Forchheimer equations.

| Category | Fracture Aperture (mm) | Fitting Equation | $R^2$ | Akaike Information Criterion (AIC) | Schwarz's Criterion (SC) | AARE |
|---|---|---|---|---|---|---|
| Linear Equation | 0.77 | −∇P = 23.0001Q | 0.9973 | 7.26 | 7.31 | 4.6% |
| | 1.18 | −∇P = 8.7935Q | 0.8793 | 10.82 | 10.87 | 26.1% |
| | 1.97 | −∇P = 2.7505Q | 0.8542 | 9.99 | 10.04 | 31.5% |
| | 2.73 | −∇P = 1.0793Q | 0.8871 | 7.86 | 7.90 | 25.6% |
| Forchheimer Equation | 0.77 | −∇P = 21.1457Q + 0.0927Q² | 0.9992 | 6.20 | 6.30 | 2.2% |
| | 1.18 | −∇P = 1.7394Q + 0.1761Q² | 0.9957 | 7.61 | 7.71 | 9.1% |
| | 1.97 | −∇P = 0.4776Q + 0.0329Q² | 0.9997 | 4.03 | 4.13 | 1.1% |
| | 2.73 | −∇P = 0.3109Q + 0.0098Q² | 0.9976 | 4.18 | 4.26 | 3.0% |

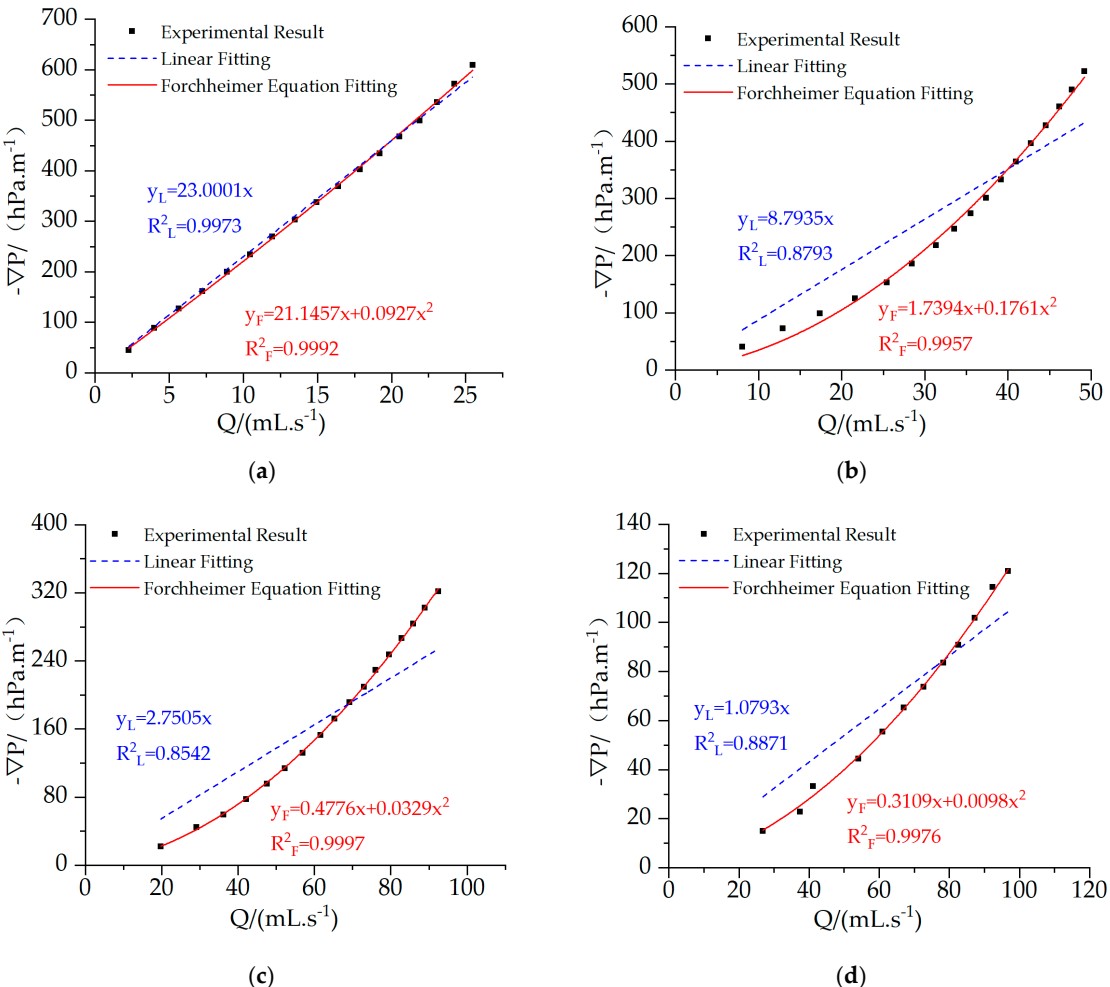

**Figure 3.** Fitted curves of $-\nabla P$ with Q by linear and Forchheimer's equations for: (**a**) e = 0.77 mm; (**b**) e = 1.18 mm; (**c**) e = 1.97 mm; and (**d**) e = 2.73 mm.

All of the scatterplots of the volume of flow and the pressure gradient are similar to a parabola except at e = 0.77 mm, which is in accordance with the results that reported by Shu et al. [22], Wang et al. [35], and Akinbodewa et al. [36]. They show obvious non-Darcian flow in the test, since the shape of the line is not linear but parabolic. The transition from linear to nonlinear behavior is like that described by Andrade et al. [37].

The linear equation does not perform well when used to fit the relationship between the volume of flow and the pressure gradient except at e = 0.77 mm (Absolute Average Relative Error, $AARE_L$ = 4.6–31.5%), while the Forchheimer equation fits well with larger determination coefficients ($AARE_F$ = 1.1–9.1%). The sum of squares of residual errors of the Forchheimer equation was one to three orders of magnitude smaller than that of the linear Equation and has smaller values of Akaike information criterion (AIC) and Schwarz's criterion (SC). Thus, the Forchheimer equation is more suitable for describing the law of flow movement in fissures.

### 3.2. Comparison between Experiment and Simulation

The realizable k–$\varepsilon$ model was used to calculate the pressure drop in a single fracture with different apertures. The results of the CFD model were compared with those of the laboratory test at the same fracture aperture. Figure 4 shows that all results of the CFD model were close to those of the physical test, with the determination coefficient $R^2$ ranging from 0.9643 to 0.9951. As is well-known, when the

determination coefficient $R^2$ of a non-intercepting fitting curve is close to one, the two sets of data are similar.

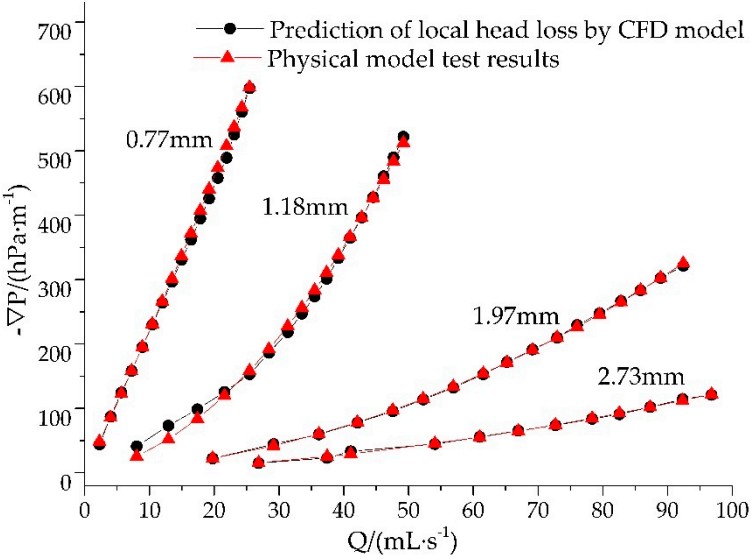

**Figure 4.** Comparison between computational fluid dynamics (CFD) modeling results and physical model test results for e = 0.77 mm, 1.18 mm, 1.97 mm, 2.73 mm.

To validate the above conclusion, Pearson's correlation coefficient was employed for further analysis. As shown in Table 3, all Pearson's correlation coefficients were close to one, indicating a strong correlation between the CDF modeling results and the numerical results, which is consistent with Figure 4. This means that the CFD model accurately simulated the flow of fracture fluid under specified conditions, consistent with the results of Wang et al. [38].

**Table 3.** Analysis using Pearson's correlation coefficient.

| Fracture Aperture (mm) | 0.77 | 1.18 | 1.97 | 2.73 |
|---|---|---|---|---|
| Pearson's correlation coefficient | 0.995 | 0.996 | 0.999 | 1 |

From the above analysis, it can be inferred that the CFD model was useful for studying the characteristics of flow in fractures, and it has potential in flexural crack modelling.

### 3.3. Effect of Fracture Shape on Head Loss

It is known that the local head loss changes when the fracture aperture changes. In order to study the relationship between the fracture shape (corner θ) and the local head loss, the fracture width is should be fixed first.

Bernoulli's equation represents the conservation of mechanical energy of an ideal fluid and can represent the process of constant energy change of the fluid per unit mass, the relation of element flow on any section, as shown in Equation (5) [39]:

$$\frac{v^2}{2g}+z+\frac{P}{\varrho g}= const \tag{6}$$

Based on this, the energy equation of the real fluid with energy loss at the cross-section of upstream and downstream can be written as follows [28]:

$$\frac{v_1^2}{2g\alpha_1}+z_1+\frac{P_1}{\varrho g}=\frac{v_2^2}{2g\alpha_2}+z_2+\frac{P_2}{\varrho g}+h_1+h_f \tag{7}$$

where $v$ is flow velocity in the fracture, (m·s$^{-1}$); $g$ is gravitational acceleration, 9.8 m/s$^2$; $h_f$ is the local head loss caused by the shape of the fissure ($\theta$); $h_l$ is frictional head loss; "1" and "2" are cross-sectional symbols of upstream and downstream, respectively, and $\alpha$ is the total flow correction factor. For fully developed turbulent flow, $\alpha \approx 1$.

The total head loss from upstream to downstream is set to $\Delta H$, which can be written as:

$$\Delta H = h_l + h_f \tag{8}$$

To study the local head loss caused by the shape of the fracture, a comparative numerical model was constructed with a fracture aperture of 1 mm, length of 100 mm, width of 20 mm, and the value of $\theta$ at zero. Under the same conditions, we established seven numerical models with $\theta$ = 30, 45, 60, 90, 120, 135, and 150 degrees. The contour maps of pressure at flexural crack with different $\theta$ can be seen in Figure 5.

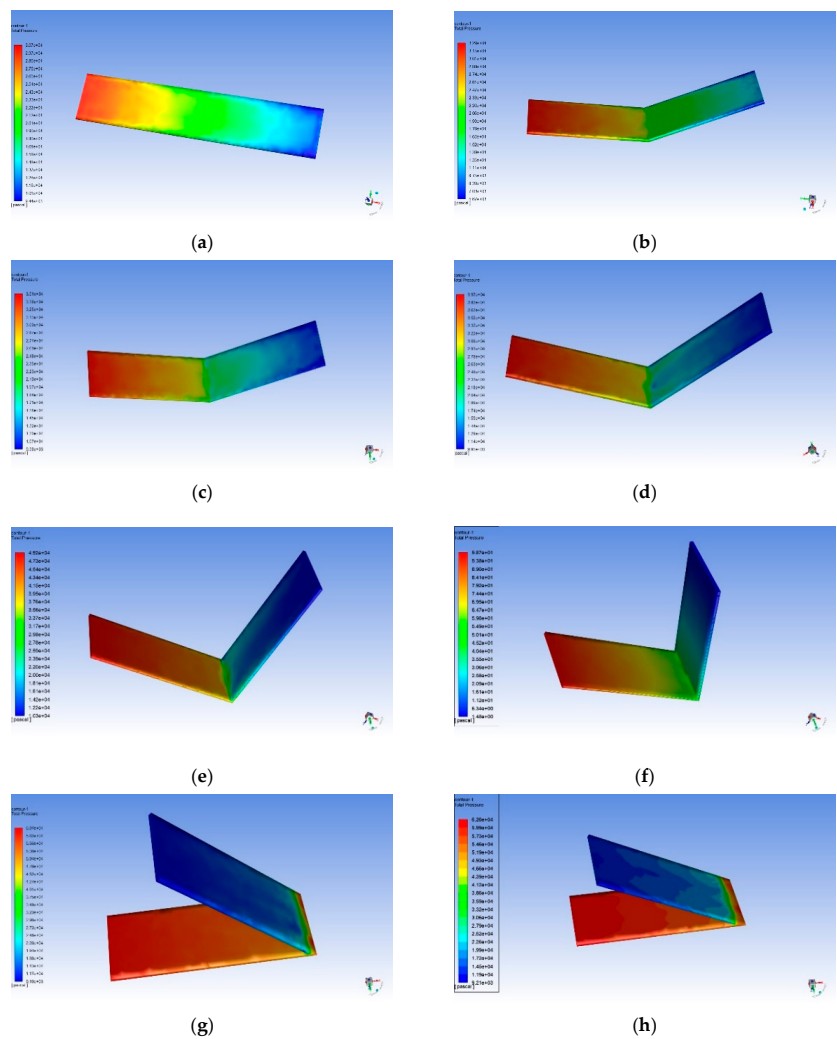

**Figure 5.** The contour maps of pressure at flexural crack with different $\theta$ (**a**) 0°; (**b**) 30°; (**c**) 45°; (**d**) 60°; (**e**) 90°; (**f**) 120°; (**g**) 135°; (**h**) 150°.

In the CFD numerical model test, except for the shape of the fracture ($\theta$), the boundary conditions and parameters of each model were the same, because of which the frictional head loss of each model was the same ($h_{l_0} = h_{l_i}$). When $\theta$ = 0, the local head loss caused by fracture shape was 0 ($h_{f_0} = 0$). By

comparing the results of these models, the local head loss caused at different values of θ at different flow velocities was calculated.

$$\Delta H_i - \Delta H_0 = h_{fi} \tag{9}$$

where $\Delta H_0$ is the head loss when θ = 0, and $\Delta H_i$ is that when θ = i.

The local head loss of each flexural crack at different angles (θ) is shown in Figure 6. It is clear that local head loss caused by fracture shape increased with the flow velocity and the crossed angle.

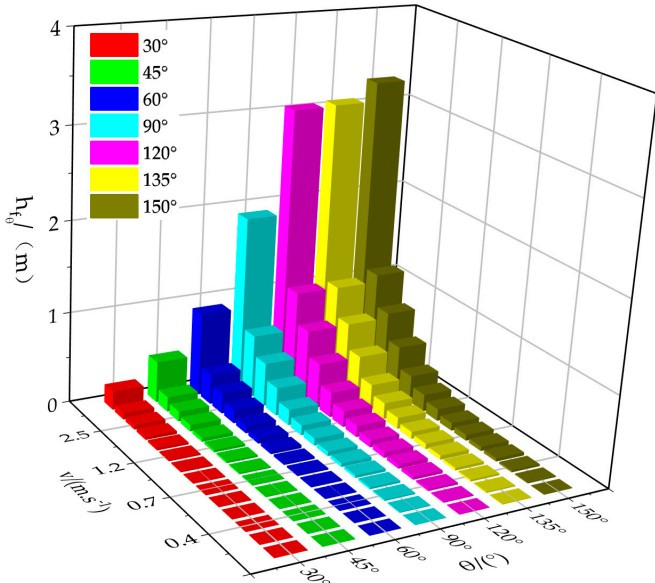

**Figure 6.** The local head loss at different values of θ. The *x*-axis is θ, the *y*-axis is flow velocity, and the *z*-axis is local head loss.

The velocity distribution in the water-crossing section is more uneven in non-Darcian flow, and there is shear stress between the layers of flow [40]. When the fluid flowed through the turning point of the flexural crack in a state of turbulence where the inertial force played a dominant role, the flow could not change direction abruptly along the fracture shape, with the consequence that the mainstream of fluids was separated from the side-wall of the fracture, and vortices were generated [41]. The eddy current and mainstream were superimposed, and spiral motion occurred downstream of the corner. In this motion, the generation, collision, merging, splitting, and disappearance of the eddy current consumed a large amount of the fluid's mechanical energy, which was the local head loss caused by the shape of the fracture.

It has been found that the larger θ is, the more active the motion of the eddy current is, and the greater the head loss is [42]. This is consistent with the results of this study. Some researchers have observed that initial pressure on the fluid controls its instantaneous flow rate in pipes. When the shape of the pipe remains constant, the local resistance loss increases with the Reynolds number [43], which agrees with the results of this study. Therefore, flow velocity and fracture shape are the important factors affecting local head loss.

### 3.4. Establishment of Fast Calculation of Local Head Loss

A pairwise correlation analysis between the independent variables (θ and *v*) and the dependent variable (local head loss $h_f$) was quantitatively represented using the Pearson coefficient, as shown in Table 4.

**Table 4.** Pairwise correlation analysis between $h_f$ and $\theta$ as well as $v$.

| | Parameter | Facture Shape ($\theta$) | Flow Velocity ($v$) |
|---|---|---|---|
| Local head loss ($h_f$) | Pearson's correlation coefficient | 0.296 | 0.758 |
| | Significant (bilateral) P | 0.003 | 0 |

From the above tables, it is clear that Pearson's correlation coefficient had the value $P_v > P_\theta$, indicating that the influence of $v$ on local head loss was greater than that of $\theta$. As the flow velocity and fracture shape are positively correlated with head loss, the calculation model, $h_f(v, \theta)$ should consider all of them, and can be expressed as [43]:

$$h_f = \xi \frac{v^2}{2g} \tag{10}$$

$\xi$ is the coefficient of local head loss that is usually related to the geometry of the member and the nature of the fluid flowing through the member, and it can be rewritten as:

$$\xi = \frac{2gh_f}{v^2} \tag{11}$$

$\xi^*$ is the average value of $\xi$ at different velocities. As shown in Figure 7, the value of $\xi^*$ is nonlinearly positively correlated with $\theta$, and error caused by changes in the flow velocity is minimal. Therefore, the calculation model for the local head loss coefficient is a function related only to the shape of the fracture but not to velocity.

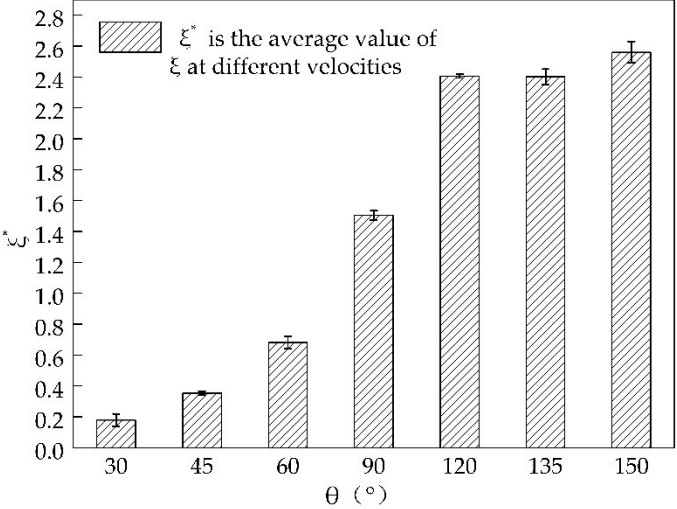

**Figure 7.** Curves of the mean numbers of $\xi^*$ with $\theta$. The error analysis line expresses error caused by the change in flow velocity.

The curve characteristic observed is S-shaped, which is somewhat similar to a logistic function. The logistic function was first used to reflect a nutritional relationship to population mathematical model in 1838, but it has been found useful in many fields. The original logistic function is:

$$y = \frac{b}{1 + Ce^{-at}} \tag{12}$$

where t is the time variable, and a, b, and C are the model constants.

The deformation of logistic equation [44] was used to fit the head loss coefficient as follows:

$$\xi(\theta) = A_2 + \frac{A_1 - A_2}{1 + \left(\frac{\theta}{b}\right)^p} \tag{13}$$

The crossed angle ($\theta$) was the independent variable with a range of 0–180 degrees. $A_1$, $A_2$, b, and $p$ are the characteristic parameters of the equation. For our study, they were $A_1 = 0.1673$, $A_2 = 3.7346$, b = 88.0791, and $p = 3.7179$.

Based on the Equation (12), we can quickly calculate the local head loss:

$$h_f(\theta, v) = \frac{v^2}{2g}\left(3.7346 + \frac{0.1673 - 3.7346}{1 + (\theta/88.0791)^{3.7179}}\right) \tag{14}$$

where $g$ is the gravity coefficient, with the value of 9.8 m/s$^2$.

### 3.5. The Use of Fast Calculation Equation of Head Loss

During fluid flow, the head loss includes the sum of the frictional head loss and the local head loss listed in Equation (7), while in Section 3.1, we found that "∇P-Q" can be well characterized by the Forchheimer equation. Therefore, the frictional head loss can be calculated according to the Forchheimer equation obtained previously. From numeric simulation, the fitting Forchheimer equation between hydraulic gradient ($J$) and flow velocity with fracture aperture of 1 mm and transverse width of 2 cm was obtained, with $R^2 > 0.99$, as follows:

$$J = 0.6803v + 1.1035v^2 \tag{15}$$

Considering the distance of frictional head loss, the calculation equation of frictional head loss can be obtained as:

$$h_l = \left(0.6803v + 1.1035v^2\right) \times L \tag{16}$$

where $L$ is the length of the fracture.

The above quantitative equations (Equations (13) and (15)) can be applied to the calculation of head loss when the crack with aperture is 1 mm and transverse width is 2 cm. To apply this, a network fissure with four nodes 1#–4# was set as shown in Figure 8.

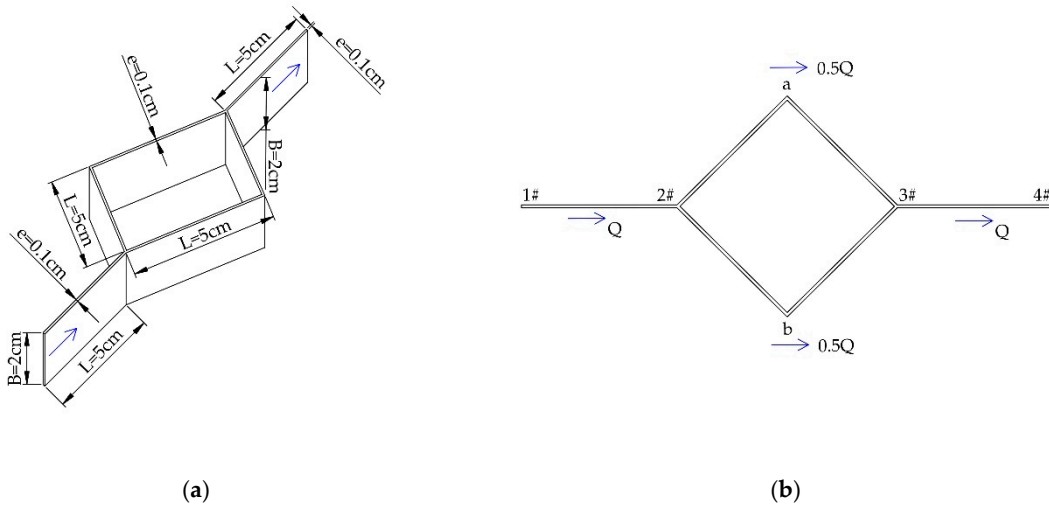

(**a**)            (**b**)

**Figure 8.** Model of an assumed sparse fracture network. (**a**) Structure of the fracture network. (**b**) Movement path of fluid in fracture network.

According to Kirchhoff's law, the flow equation at the node was established:

$$Q_{1-2} = Q_{2-a-3} + Q_{2-b-3} = Q_{3-4} \tag{17}$$

Assuming that the flow velocity at the entrance (1#) and exit (4#) is $v$, the flow velocity in each flow path is:

$$v_{1-2} = 2v_{2-a-3} = 2v_{2-b-3} = v_{3-4} = v \tag{18}$$

In Figure 7, We can divide the network fracture into multiple single fractures and local loss occurrence points, so the head loss of the fracture network is the sum of them:

$$\Delta H_{1\text{-}4} = h_{l(1\text{-}4)} + h_{f(1\text{-}4)} \tag{19}$$

where:

$$h_{l(1\text{-}4)} = h_{l(1\text{-}2)} + h_{l(2\text{-}a)} + h_{l(a\text{-}3)} + h_{l(2\text{-}b)} + h_{l(b\text{-}3)} + h_{l(3\text{-}4)} \tag{20}$$

$$h_{f(1\text{-}4)} = h_{f(1\text{-}2\text{-}a)} + h_{f(2\text{-}a\text{-}3)} + h_{f(a\text{-}3\text{-}4)} + h_{f(1\text{-}2\text{-}b)} + h_{f(2\text{-}b\text{-}3)} + h_{f(b\text{-}3\text{-}4)} \tag{21}$$

Among the Equations (19) and (20), $h_{l(1\text{-}2)}$, $h_{l(2\text{-}a)}$, $h_{l(a\text{-}3)}$, $h_{l(2\text{-}b)}$, $h_{l(b\text{-}3)}$, and $h_{l(3\text{-}4)}$ refer to the frictional head loss of the fracture segment of (1-2), (2-a), (a-3), (2-b), (b-3), and (3-4); $h_{f(1\text{-}2\text{-}a)}$, $h_{f(2\text{-}a\text{-}3)}$, $h_{f(a\text{-}3\text{-}4)}$, $h_{f(1\text{-}2\text{-}b)}$, $h_{f(2\text{-}b\text{-}3)}$, and $h_{f(b\text{-}3\text{-}4)}$ refer to the local head loss at the positions where the direction of flow changes. Parameters of each fracture or intersection point are listed in Tables 5 and 6.

**Table 5.** Parameters of each fracture for frictional head loss calculation.

| Fracture | Flow Velocity (m/s) | Length $L$ $(v)$ (m) | Frictional Head Loss (m) | Sum of Frictional Head Loss (m) |
|---|---|---|---|---|
| $h_{l(1\text{-}2)}$ | $v$ | 0.05 | $0.05\,(av + bv^2)$ | |
| $h_{l(2\text{-}a)}$ | $0.5v$ | 0.05 | $0.05\,(0.5av + 0.25bv^2)$ | $0.2av + 0.15bv^2$ ($a = 0.6803$, $b = 1.1035$) |
| $h_{l(a\text{-}3)}$ | $0.5v$ | 0.05 | $0.05\,(0.5av + 0.25bv^2)$ | |
| $h_{l(2\text{-}b)}$ | $0.5v$ | 0.05 | $0.05\,(0.5av + 0.25bv^2)$ | |
| $h_{l(b\text{-}3)}$ | $0.5v$ | 0.05 | $0.05\,(0.5av + 0.25bv^2)$ | |
| $h_{l(3\text{-}4)}$ | $v$ | 0.05 | $0.05\,(av + bv^2)$ | |

**Table 6.** Parameters of each intersection point for local head loss calculation.

| Intersection Point | Flow Velocity (m/s) | Bending Angle θ (°) | ξ | Sum of Local Head Loss (m) |
|---|---|---|---|---|
| $h_{f(1\text{-}2\text{-}a)}$ | $0.5v$ | 45 | 0.4387 | |
| $h_{f(2\text{-}a\text{-}3)}$ | $0.5v$ | 90 | 2.0224 | |
| $h_{f(a\text{-}3\text{-}4)}$ | $0.5v$ | 45 | 0.4387 | $0.7250v^2/g$ |
| $h_{f(1\text{-}2\text{-}b)}$ | $0.5v$ | 45 | 0.4387 | |
| $h_{f(2\text{-}b\text{-}3)}$ | $0.5v$ | 90 | 2.0224 | |
| $h_{f(b\text{-}3\text{-}4)}$ | $0.5v$ | 45 | 0.4387 | |

The head losses calculated are listed in Table 7.

**Table 7.** Total head loss calculation of network fracture.

| Flow Velocity (m/s) | 0.2 | 0.4 | 0.8 | 1.2 | 1.5 | 2 | 3 | 5 |
|---|---|---|---|---|---|---|---|---|
| Frictional head loss (m) | 0.0338 | 0.0809 | 0.2148 | 0.4016 | 0.5765 | 0.9342 | 1.8979 | 4.8184 |
| Local head loss (m) | 0.0030 | 0.0118 | 0.0473 | 0.1065 | 0.1664 | 0.2959 | 0.6658 | 1.8494 |
| Total head loss (m) | 0.0368 | 0.0927 | 0.2621 | 0.5082 | 0.7430 | 1.2301 | 2.5637 | 6.6678 |

From Table 7, it can be seen that local head loss accounts for a small proportion when the flow velocity is small, but it grows fast as the flow velocity increases because local head loss is proportional to the square of speed. Water inrush from the underground projects, such as tunnels, is always very fast, so the local head loss is necessary to be taken into consideration when doing water inflow predication. In addition, parameters of Equations (13) and (15) need to be recalibrated before use because they are obtained from the specific conditions when the fracture aperture is 1 mm and its width is 20 mm.

## 4. Conclusions

- Based on the physical experiment, non-Darcian characteristics were found when the speed was high enough in fracture. The Forchheimer equation fits the experimental data well between pressure gradient and volume of flow. The fracture width and fluid velocity, tested in the physical experiment, were important factors affecting the fluid flow pattern.
- The non-Darcian flow could be well represented by CFD model in a single fracture. It was employed to simulate the behavior of flow in flexural crack and obtain the result that local head loss was closely related to flow velocity and fracture shape, which can be represented as $\Delta H(\theta, v)$.
- The local head loss coefficient is almost independent of speed, while it has a very strong relationship with the bending angle $\theta$ of flexural crack. It can be treated as a logistic function of $\theta$, such as $\zeta(\theta)$.
- Friction head loss and local head loss can be calculated separately when the fracture system is complicated, but it is strongly recommended to take the local head loss into consideration when the flow speed is high, as it accounts for a large proportion of the total head loss.

**Author Contributions:** Investigation, L.H.; Investigation, P.L.; Data curation, K.S.; Mapping, C.M.; Writing—review & Software, J.L.; Writing—original draft & editing, X.B. All authors have read and agreed to the published version of the manuscript.

**Funding:** This research was funded by the National Science Foundation of China, grant number [No.41602241, U1734205] and the major subject of science and technology of Sichuan province, grant number [No.2019YFS0509].

**Acknowledgments:** This research was partly supported by software of China Ship Scientific Research Center.

**Conflicts of Interest:** The authors declare no conflict of interest. The funding sponsors had no role in the design of the study; in the collection, analyses, or interpretation of data; in the writing of the manuscript, or in the decision to publish the results.

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
