# Peer review of "A Fast Calculation Model for Local Head Loss of Non-Darcian Flow in Flexural Crack"

_water, doi:10.3390/w12010232_

Round 1
Reviewer 1 Report
The work address the calculation of head loss of Non-Darcian flow in flexural cracks with different shapes to predict the energy loss. Laboratory set-up is established to do the experiment and authors indicate that these results are validated using CFD. The topic and the methodology could be interesting for the reader, but several aspects must be clarified and improved. The paper has very important scientific and methodological shortcomings and it must be rewritten.
General comments.
Authors indicate that a validation based-on CFD model is applied. Even indicate in the conclusion section the next: “the CFD is useful for studying the characteristic s of flow in fracture….” However, few aspects are explained about this specific simulation addressed, being this an essential issue of the raised research because depending on the validity of the based on CFD methodology applied, the reliability of the methods used could be assessed. The complete description of the simulation using CFD must be included and justified in the paper, indicating the software used, the specific meshing process followed, type of mesh, boundary conditions, assumed calculation assumptions, all the simulation features, geometry generation, etc and preferably images and graphs extracted from CFD analysis should be displayed. Otherwise, the article lacks the necessary scientific rigour.
Specific comments.
Eq (1) really refers to the fit of the pressure gradient fitted with respect the flow rate. Based on it, authors obtain the correlation of data in the laboratory context. It could be adequate but, in this case, methodology is designed based on literature and this should be better highlighted in the manuscript.
Figure 1 is unintuitive. It is not clear. What is the position of “5” with respect the rest of devices of the experimental set-up? Figure must be improved.
Authors indicate that volumetric method is used to measure the flow rate (line 96). I think that this method consists on the direct measurement of the time to fill a container, being known the volume. For this type of measures, high errors can be made due to possible fluctuations of the flow which is not always homogeneous and stationary. The volume of the container used for the flow measuring should be indicated and whether the measurement is taken from the empty container or between two fluid levels in the container.
All the equations should be mentioned in the text. A lot of equations are indicated without connection with the manuscript (e.g. eq.4-13). The equations should be used to substantiate the description of the methodologies or the theory. The same happens with the description of each variable of each equation.
Authors indicate in the line 201 “…the process of constant energy change of the fluid per unit mass as shown in equation (5)”. This is not correct. The eq 5. Really represent the energy of the fluid per weight mass (because it is expressed in head terms (Nm/N)). That is, the total energy per unit of weight of the fluid passing through a section, which remains constant if the energy equation is fulfilled.
The difference between H (capital letter) and h must be better explained (eq 17,18 and other). That notation is unintuitive and difficult for the reader to interpret. Also the subindex of H should be better explained.
Figure 8 a) should be improved.
In eq. 28, why the total head loss is calculated as the sum of the head loss of each parallel pipe (fig. 8)? The pressure drop is accumulative, the total loss between a and b will depend on a single flow path. Furthermore, in parallel conductions, the head should be the same. It should be deeply clarified. Furthermore, 3.4 subsection in general terms is very confusing and poorly explained and not very rigorous.
Table 5 is confusing. I suggest separating the content in two table: one for the frictional head loss and another for the local head loss.
In eq. 23, when the numerical values are considered, has the gravity coefficient value been considered?
Conclusion section are very short, and it does not specify the actual contribution of the paper.
English should be thoroughly reviewed throughout the paper.
Details about the CFD simulation are barely given. It seems like the authors want to mention it without going into it and that cannot be because the interest of the research depends on those results being explained and shown clearly.
Without knowing the methodology and the results of the simulation in a clear and well presented way, the contribution cannot be valued because the interest is in the comparison of the experimental results with respecto the simulation results from CFD. In addition, the paper has scientific shortcomings that need to be corrected. I suggest the rejection of the work, but that it can be recovered if the indicated points are specified.
Author Response
Eq (1) really refers to the fit of the pressure gradient fitted with respect the flow rate. Based on it, authors obtain the correlation of data in the laboratory context. It could be adequate but, in this case, methodology is designed based on literature and this should be better highlighted in the manuscript
Reply: Thank you for your valuable comment and suggestion. In 2.3.2, the type of CFD model was first introduced, then the origin and basic equation. According to the revision suggestion, the description of the application method of CFD model was added. The CFD model was built by the software of ANSYS FLUENT. Scholars are reported to study fracture fluid by the CFD model [24-28].According to the suggestion, that the methodology is designed based on literature was explained in lines136-144. And all values of these parameters were obtained according to the recommendations by Launder et al. [33] and later experimental verification [34].
Figure 1 is unintuitive. It is not clear. What is the position of “5” with respect the rest of devices of the experimental set-up? Figure must be improved.
Reply: Thank you for your valuable comment. The figure 1(a) and figure 1(b) were improved. The test device diagram of position 5 was supplemented in figure 1(a) and figure 1(b). And we made the number of physical diagram in Figure 1(a) correspond to the number in Figure 1(b).
Authors indicate that volumetric method is used to measure the flow rate (line 96). I think that this method consists on the direct measurement of the time to fill a container, being known the volume. For this type of measures, high errors can be made due to possible fluctuations of the flow which is not always homogeneous and stationary. The volume of the container used for the flow measuring should be indicated and whether the measurement is taken from the empty container or between two fluid levels in the container.
Reply: Thank you for your valuable comment. Measuring the flow rate per unit time was considered. Because of the turbulent fluid, the flow rate data measured by rotameter in a short period time had great fluctuation and real-time performance, which was difficult to summarize the research and analysis of fluid characteristics. Therefore, the method of measuring the weight of the fluid in a certain period time and divide it by the density of the fluid to calculate the instantaneous flow of the fluid were adopted. The density of the fluid was 0.999525 at 12 ℃. In terms of improving the accuracy, the stability of the flow state in the fracture after each change of the head height was achieved in a certain time which is long enough. When collecting the output water, the time was measured by a stopwatch, while a 1000 ml beaker was used to collect 20s of the outlet fluid. Finally, the collected water was weighed by electronic balance. In order to reduce the operation error, we weighed several times continuously to ensure that the relative error of the three measurement results is less than 5%.
All the equations should be mentioned in the text. A lot of equations are indicated without connection with the manuscript (e.g. eq.4-13). The equations should be used to substantiate the description of the methodologies or the theory. The same happens with the description of each variable of each equation.
Reply: Thanks for your great suggestion. We agree with you very much. According to the suggestion, several parameter calculation formulas (eq 4-13) were deleted other than the basic algorithm, and the parameter settings in the model were listed in lines 144-157.
Authors indicate in the line 201 “…the process of constant energy change of the fluid per unit mass as shown in equation (5)”. This is not correct. The eq 5. Really represent the energy of the fluid per weight mass (because it is expressed in head terms (Nm/N)). That is, the total energy per unit of weight of the fluid passing through a section, which remains constant if the energy equation is fulfilled.
Reply: Thanks for your valuable questions. In this regard, Equation 5 is the conservation equation of the mechanical energy of the ideal fluid. We assure that "kinetic energy + gravitational potential energy + pressure potential energy = constant". In this paper, the “Etm“ was changed to ”constant“ to avoid misunderstanding of formula and default unit. Meanwhile, the inevitable energy loss caused by the movement of the fluid in the fracture cannot be ignored as the motion state of the actual fluid was investigated. Hence, the real fluid with energy loss could be obtained by combining the N-S equation and the energy conservation equation, which is called the Bernoulli equation , written as equation (6). Published books like < Fundamentals of fluid mechanics> and both show agreements of the Bernoulli equation.
The difference between H (capital letter) and h must be better explained (eq 17,18 and other). That notation is unintuitive and difficult for the reader to interpret. Also the subindex of H should be better explained.
Reply: Thanks for your valuable questions. The question was clarified in lines of 209-213. The H is the head loss, which is the sum of the local head loss and the head loss along the way. The ΔH was defined as the head loss from upstream to downstream. The ΔHi was defined as the head loss when θ equaled i, while ΔH0 was the head loss when θ equaled 0. Therefore, the Equation 8 and Equations 18-21 were optimized.
In eq. 28, why the total head loss is calculated as the sum of the head loss of each parallel pipe (fig. 8)? The pressure drop is accumulative, the total loss between a and b will depend on a single flow path. Furthermore, in parallel conductions, the head should be the same. It should be deeply clarified. Furthermore, 3.4 subsection in general terms is very confusing and poorly explained and not very rigorous.
Reply: Thank you for your valuable comment. I mistakenly wrote the chapter number. According to this opinion, I reorganized the chapter 3.5 the use of fast calculation equation of local loss. And the opinion was responded as follows:
This research was based on the fact of that the pressure drop was accumulative and the total loss between a and b depended on a single flow path. Therefore, when calculating the network fracture head loss, a single fracture model with fracture width of 1mm and length of 100mm was first established by the CFD model to measure the resistance loss data along the fracture when the velocity range was 0.2-5m/s. Afterwards, the Forchheimer equation was adopted to express the pressure drop in the velocity range, which resulted in R2>0.99 of equation 14. The hydraulic loss efficiency caused by inertial force and viscous force in the process of fracture movement of high-speed non-Darcian fluid were generalized by the Forchheimer equation, which had strong structural stability reflecting physical phenomena and could express the pressure drop of head loss along the path within a certain test scope. We learned it from many books like and . Therefore, the head loss along the network fracture was calculated by equation 14. In the setting of the network fracture model, the fracture conditions and fluid characteristics were the same as those of the single fracture model, only the velocity was taken as the variable. So, the empirical value -▽P and the runoff distance L could be multiplied to get the frictional head loss during the distance. In addition, the velocity at the inlet and outlet of the network fracture were same, as well as, the instantaneous flow of the fluid through 2#-a-3# were same as that of 2#-b-3#, and their sum meant the instantaneous flow of inlet and outlet. As a result, the velocity in the fracture 2#-a-3# and 2#-b-3#, were half of Inlet and outlet flow rate. Based on aboves, the calculation of hydraulic loss of network fracture in this paper was studied.
Table 5 is confusing. I suggest separating the content in two table: one for the frictional head loss and another for the local head loss.
Reply: Thanks for your suggestion. According to the proposal, we divided table 5 into two parts, which were shown in Table 5 and Table 6.
In eq. 23, when the numerical values are considered, has the gravity coefficient value been considered?
Reply: Thanks for your valuable questions. The gravity coefficient was considered in this equation. When brought the gravity into the calculation, only velocity was the variable in the model.
Conclusion section are very short, and it does not specify the actual contribution of the paper.
Reply: Thanks for your great suggestion. According to the revision suggestion, we rewrote the conclusion. Thus emphasizing the actual contribution of this paper,the detail change as fellow:
Based on the physical experiment, the non-Darcian movement characteristics could be described by the Forchheimer equation in the high-speed fracturing fluid flow. The fracture width and fluid velocity, which were tested in the physical experiment, were important factors affecting the fluid flow pattern.
The non-Darcian flow could be well represented by CFD model in a single natural fracture. At the same time, the numerical model established by the CFD to could strictly control the test conditions. Moreover, it could simulate the physical test models which were hard to implement. The CFD model could be used for further study of flow characteristics in fractures.
The local head loss was closely related to flow velocity and fracture shape and can be represented as. The relationship between the local head loss coefficient ζ (θ) and the crack shape θ can be effectively summarized with the logistic function, shown in equation (13).
The friction resistance loss from the local resistance loss could be separated by the head loss calculation of network cracks. The calculation process of local head loss can be optimized by the fast calculation model composed of logistic function.
English should be thoroughly reviewed throughout the paper.
Reply: Thanks for your great suggestion. We have proofread the manuscript, to reduce the typos.
Details about the CFD simulation are barely given.
Reply: Thanks for your suggestion. We added the parameter setting of the CFD model in lines 145-158. If necessary, we can also provide screenshots of the software settings. In addition, in Figure 6 and Figure 7, the values analyzed were the operation values of the CFD model.
With kind regards,
Jian Liu & Xue Bai

Reviewer 2 Report
Manuscript Number: water-662588
Title: A fast calculation model for local head loss of non-Darcian flow in flexural crack
The current manuscript aims to develop a model to predict the head loss – fracture, taking in consideration the aperture and shape of the fracture. Furthermore, the authors built
CFD models of flexural crack with different angles to compute the energy loss as well as comparison was presented. Overall the manuscript contains a well thought out. The conclusion is well explained by the presented results. I have comments below that should be addressed:
Q1. The abstract has to be rewritten to has a certain level of quantification in terms of key findings.
Q2. Since, there are many factors affecting the non-Darcian flow in flexural crack and consequently the head loss. Why, the authors picked only the aperture and shape of the fracture in their model. Is this the only model that addressed the effect of these parameters…that was not clear in the manuscript. Could you please clarify that in the manuscript? I do suggest to discuss and add that in the lines of 56-60.
Q3. The physical picture of the test device in Fig. 2 is not clear. I do suggest to add labels to the image, to make it clear to the reader. Also, it does not has all the components of the system.
Q4. What do you mean with the “width measurement system” (line 68)?
Q5. Is the “water intake system” a flowmeter or what? Please clarify and add the part numbers of it in the revised manuscript. What is the volumetric method? Is it rotameter or what? And how accurate is it?
Q6. What is the reference behind picking the following width: e=0.77 mm, e=1.18 mm, e=1.97 mm, and e=2.73 mm. Simply, why the authors picked that values?
Q7. Uncertainty quantifications are highly recommended for the current indoor experiments? On the light of the current lab experiments. I do think that there is a higher level of uncertainty?
Q8. Did all the experiments execute at the same operating temperature of the water?
Q9. Did the authors consider the development of the fluid flow, hydrodynamic boundary layer development, before entering the system? Simply, the length of the tube from he water intake to the fissure system is long enough to ensure fully developed flow to the fissure/test section?
Q10. I do suggest to use AARE, Average Absolute Relative Error, to quantify the differences between the linear fitting, experiments, and Forchheimer Equation Fitting.
Q11. Proofreading is required for the manuscript as there are few typos in the manuscript.
Q12. Provide the funding number, if available, to the acknowledgement (line 343).
Author Response
Dear reviewer,
Thank you for your comments on our article. The revisions are marked in the manuscript and listed to be responded separately. We are available to provide responses if there are still some questions. Thank you very much for your suggestions and consideration.
Response:
The abstract has to be rewritten to has a certain level of quantification in terms of key findings.
Reply: Thank you for your valuable comment. According to the revision suggestion, we rewrote the abstract, thus emphasizing the research purpose and having a certain level of quantification in terms of key findings. Now we clearly know that the Forchheimer equation can describe the fluid characteristics effectively, which is, the CFD model is very suitable to simulate the physical model, and the fast calculation model is of great value to the calculation of local resistance loss. The summary is as follows:
This study proposed a fast calculation model that the aperture and the shape of flexural crack are considered to predict local head loss directly. The fracture fluid represented characteristics of non-Darcian flow that could be depicted by the Forchheimer equation when the flow velocity was sufficiently large in a physical experiment on the single fracture using fracture apertures e of 0.77, 1.18, 1.97, and 2.73 mm (R2 > 0.99). Following the formulation of a numerical model from computational fluid dynamics (CFD) which is to verify the calculation of single fractures, a CFD model of flexural crack with different angles were built to compute the energy loss of each, which can verify the physical experiment results very well (Pearson correlation coefficient >0.99). After eliminating the influence of crack width, it is found that the local head loss of the flexural crack varied with the bending angle, and its coefficient was expressed by the deformation of the logistic equation. Therefore, we established a fast calculation model with fracture angle and velocity as variables. By using this model, as well as a frictional head loss equation fitted by the Forchheimer equation (R2 > 0.99), the head loss of crossed fissures with fixed fracture aperture could be easily calculated.
Since, there are many factors affecting the non-Darcian flow in flexural crack and consequently the head loss. Why, the authors picked only the aperture and shape of the fracture in their model. Is this the only model that addressed the effect of these parameters…that was not clear in the manuscript. Could you please clarify that in the manuscript? I do suggest to discuss and add that in the lines of 56-60.
Reply: Thank you for your valuable comment. The question is clarified in lines of 59-67: This study proposed a fast calculation model that the aperture and the shape of flexural crack are considered to predict head loss directly by using an indoor physical experiment and numerical verification. It is difficult to consider all factors when using the Forchheimer equation to model flow in fractures. In order to investigate the influence of each factor on fracture fluid, the synergistic effects of each conditions need to be reduced and the influence rules of each conditions need to be researched separately. Fracture aperture and shape are important parameters to generalize fracture development, which explained why only the two most significant parameters were considered here. It is hoped to improve the prediction accuracy of fracture fluid motion in applications.
The physical picture of the test device in Fig. 2 is not clear. I do suggest to add labels to the image, to make it clear to the reader. Also, it does not has all the components of the system.
Reply: Thank you for your valuable comment and suggestion. According to the revision suggestion, we have supplemented the description of the test device in lines 77-79 and 87-91. Then the figure 1(a) and figure 1(b) were improved, to make the number of the physical diagram in figure 1(b) correspond to the number in figure 1(a).
What do you mean with the “width measurement system” (line 68)?.
Reply: Thank you for your valuable question. We improved the description of the system in lines82-84, then added the photo of the microscope in Figure 1(b).
Is the “water intake system” a flowmeter or what? Please clarify and add the part numbers of it in the revised manuscript. What is the volumetric method? Is it rotameter or what? And how accurate is it?
Reply: Thanks for your valuable questions. The “water intake system” is a set of water distribution device that controlled the inflow water head and ensured the stability of water inflow, by setting an overflow port on a movable and fixed water distributor. The volumetric method is to measure the instantaneous flow, by divides the weight of the collected fluid by the time of collection and the density of the fluid. The density of the fluid is 0.999525 at 12 ℃. In terms of improving the accuracy, we always waited more than 15 minutes after changing the intake water head height, for the fluid to stabilize. When measured the instantaneous flow of fluid, water was collected by 1000 ml beaker, the time of water collection was measured by stopwatch, that we always collect the output water in about 20s. The weight of collected water was weighted by electronic balance. In order to reduce the operation error, we weight several times continuously to ensure that the relative error of the three measurement results is less than 5%.
What is the reference behind picking the following width: e=0.77 mm, e=1.18 mm, e=1.97 mm, and e=2.73 mm. Simply, why the authors picked that values?
Reply: Thanks for your valuable questions. We clarified the question in lines of 102-106: When picking the fracture width, we plan to select the fracture width within the range of 0.05-0.10cm, 0.10-0,15cm, 0.15-0.2cm, 0.2-0.3cm, so as to test the fluid motion in different gradients of fracture width. After fixed the fracture width, testing the fracture width by fracture aperture measurement system, Therefore, we determine the fracture width are 0.77mm, 1.18mm, 1.97mm, 2.73mm respectively.
Uncertainty quantifications are highly recommended for the current indoor experiments? On the light of the current lab experiments. I do think that there is a higher level of uncertainty?
Reply: Thank you for your valuable comment. There are many uncertainties in the research of groundwater fracture flow patterns, and there are many reasons that affect the fluid movement. In the laboratory test, we can only reduce the test variables as much as possible, as to analyze the influencing factors one by one. In the process of uncertainty quantification, we have completed a lot of test optimization according to the actual situation. For example, the research and development of water inlet system, the selection of gap width measurement system, the selection of flow measurement system, etc. We hope that under above test conditions, the test uncertainty will be reduced. In addition, from another point of view, the characteristics of our physical experiment data are similar to those of many other scholars [5-7,22,35-37], and the uncertainty effect of physical experiment should be acceptable
Did all the experiments execute at the same operating temperature of the water?
Reply: Thanks for your valuable questions. The test was carried out in the same climate, so the temperature did not fluctuate much. The water temperature was about 12 ℃. We explain it in line 86. In addition, all water temperature parameters of the study are also 12 ℃.
Did the authors consider the development of the fluid flow, hydrodynamic boundary layer development, before entering the system? Simply, the length of the tube from the water intake to the fissure system is long enough to ensure fully developed flow to the fissure/test section?
Reply: Thanks for your valuable questions. We have been considered the development of fluid flow in experiments. We keep the distance from the intake to the first pressure measuring point of the pressure measuring system is 15 cm, that to made the water intake to the fissure system is long enough to ensure fully developed flow to the test section.
I do suggest to use AARE, Average Absolute Relative Error, to quantify the differences between the linear fitting, experiments, and Forchheimer Equation Fitting.
Reply: Thanks for your great suggestion. We added the analysis of Average Absolute Relative Error in table 2 (line 165), to quantify the differences between the linear fitting, experiments, and Forchheimer Equation Fitting.
Proofreading is required for the manuscript as there are few typos in the manuscript.
Reply: Thanks for your great suggestion. We have proofread the manuscript, to reduce the typos in the manuscript.
Provide the funding number, if available, to the acknowledgement. (line 343).
Reply: Thanks for your suggestion. We had written the funding number in line344.
With kind regards,
Jian Liu & Xue Bai

Round 2
Reviewer 1 Report
The paper has been improved in this version and some methodological aspects have been clarified. However, modifications and explanations about some aspects are still needed.
Comments
I suggest indicating the resolution of the electronic balance used to obtain the weight of the fluid.
The software version for Ansys Fluent must be indicated.
In Fig. 3, the equation for (b) is really a lineal fit (instead curve fit). Check it.
Configurations and details about the CFD simulation are important issues, so I suggest adding the screenshots of the software setting as it is indicated by the authors. Furthermore, I suggest including some screenshots about the simulation results and/or plots.
Author Response
Dear reviewer,
Thank you for your comments on our manuscript. Those comments are very helpful for revising and improving our paper, as well as the important guiding significance to others research. The revisions are marked in the manuscript and listed to be responded separately. In addition, the manuscript was made more revisions based on your two reviews. We are available to provide responses if there are still some questions. Thank you very much for your suggestions and consideration.
Response:
I suggest indicating the resolution of the electronic balance used to obtain the weight of the fluid.
Reply: Thank you for your valuable suggestion. The accuracy of the electronic balance was indicated on line 105, that is 0.01g.
The software version for Ansys Fluent must be indicated.
Reply: Thank you for your valuable comment. The software version for Ansys Fluent was indicated on line 145, which is ANSYS FLUENT 19.1.
In Fig. 3, the equation for (b) is really a lineal fit (instead curve fit). Check it.
Reply: Thank you for your valuable comment. Due to our negligence, the fitting formula of Figure 3 (b) was wrongly written. It has been modified now.
Configurations and details about the CFD simulation are important issues, so I suggest adding the screenshots of the software setting as it is indicated by the authors. Furthermore, I suggest including some screenshots about the simulation results and/or plots.
Reply: Thanks for your great suggestion. The important parameters were listed in the manuscript, and some of the screenshot of parameter setting on Fluent was list below.
Besides, according your great suggestion, the screenshot about the simulation results was added in Figure 5, that is the contour map of pressure at flexural crack with different θ.
Figure 5. The contour map of pressure at flexural crack with different θ (a) 0°; (b) 30°; (c) 45°; (d) 60°; (e)90°; (f)120°; (g) 135°;(h) 150°.
With kind regards,
Jian Liu & Xue Bai

Reviewer 2 Report
Authors have implemented all my comments.
Author Response
Dear reviewer,
Thank you for your comments on our manuscript. All of your comments are very helpful for revising and improving our paper, as well as the important guiding significance to others research. Thank you very much for your suggestions and consideration before.
With kind regards,
Jian Liu & Xue Bai